Effects of Lecanicillium lecanii strain JMC-01 on the physiology, biochemistry, and mortality of Bemisia tabaci Q-biotype nymphs

Xie Ting
Jiang Ling
Li Jianshe
Hong Bo
Wang Xinpu
Jia Yanxia helenjia_2006@nxu.edu.cn
School of Agriculture, Ningxia University , Yinchuan, Ningxia , China
Janes Jasmine
Electronic publication date: 2019 Sep 16
Publication date: 2019
Volume: 7
Electronic Location ID: e7690
Received 2019 Feb 9; Accepted 2019 Aug 19
Copyright: © 2019 Xie et al.
Copyright year: 2019
Copyright holder: Xie et al.
License: This is an open access article distributed under the terms of the Creative Commons Attribution License, which permits unrestricted use, distribution, reproduction and adaptation in any medium and for any purpose provided that it is properly attributed. For attribution, the original author(s), title, publication source (PeerJ) and either DOI or URL of the article must be cited.
License URL: https://creativecommons.org/licenses/by/4.0/

Keywords: Lecanicillium lecanii JMC-01, Bemisia tabaci, Protective enzymes, Detoxification enzymes, Physiological and biochemical metabolism, Mortality

Funding: Ningxia 13th Five-Year Plan for Key Research and Development Program 2016BZ09-03 and 2018BBF02021-02 Natural Science Foundation of Ningxia Province 2018AAC03038 This research was funded by the Ningxia 13th Five-Year Plan for Key Research and Development Program (No. 2016BZ09-03 and No. 2018BBF02021-02) and the Natural Science Foundation of Ningxia Province (No. 2018AAC03038). The funders had no role in study design, data collection and analysis, decision to publish, or preparation of the manuscript.

==============================
Background

Lecanicillium lecanii is an entomopathogenic fungi, which was isolated from insects suffering from disease. Now, it is an effective bio-control resource that can control agricultural pests such as whitefly and aphids. There are many studies on the control of various agricultural pests by L. lecanii, but no report on its control of Bemisia tabaci biotype-Q exists. In this work, we studied the susceptibility of B. tabaci Q-biotype (from Ningxia, China) to L. lecanii JMC-01 in terms of nymph mortality and the changes in detoxifying protective enzymes activities.

Methods

B. tabaci nymphs were exposed to L. lecanii JMC-01 conidia by immersion with the host culture. Mortality was assessed daily for all nymph stages. The detoxifying and protective enzyme activity changes, weight changes, and fat, and water contents of the nymphs were determined spectrophotometrically.

Results

All instars of B. tabaci died after being infested with 1 × 108 conidia/mL. The 2nd-instar nymphs were the most susceptible, followed by the 3rd-instar nymphs. The corrected cumulative mortality of the 2nd- and 3rd-instar nymphs was 82.22% and 75.55%, respectively. The levels of detoxifying and protective enzymes initially increased and then decreased. The highest activities of carboxylesterase, acetylcholinesterase, peroxidase, and catalase occurred on the 3rd day, reaching 10.5, 0.32, 20, and 6.3 U/mg prot, respectively. These levels were 2.2-, 4.3-, 2.4-, and 1.4-fold the control levels, respectively. The highest activities of glutathione-S transferase and superoxide dismutase on the 2nd day were, respectively, 64 and 43.5 U/mg prot. These levels were, respectively, 2.7 and 1.1-fold that of the control level. The water and fat content in the infected B. tabaci nymphs decreased and differed significantly from the control levels. The weight increased continuously in the first 24 h, decreasing thereafter. At 72 h, the infestation level was about 0.78-fold that of the control level.

Conclusions

The studied L. lecanii JMC-01 strain is pathogenic to the B. tabaci Q-biotype. This strain interferes with the normal functioning of detoxifying and protective enzymes, and is also involved in the disruption of normal physiological metabolism in B. tabaci.

Introduction

The whitefly or tobacco whitefly Bemisia tabaci (Gennadius) (Hemiptera: Aleyrodidae) is a cosmopolitan insect pest with more than 900 documented host plant species. This species is considered to belong to a cryptic species complex with more than 40 morphotypes distributed across the biotypes, with the B- and Q-biotypes being the most important (Tang et al., 2018). The whitefly is of economic importance due to its direct (by sapping plant fluids and vectoring plant pathogens) and indirect (phytosanitary and quarantine measures) damage to crops (De Barro, 2011; Xu et al., 2014). Its control mainly relies on chemical pesticide application, which has resulted in the development of insecticide resistance. In addition to the emergence of resistant strains, farms and other stakeholders are challenged by safety concerns. Pesticide application causes environmental pollution, alters the abundance of natural enemies, increases pest resistance, and promotes secondary pest population resurgence (Liu et al., 2009). Environmentally-friendly pest management methods such as biological control using natural enemies and entomopathogen microorganisms (bacteria, fungi, and viruses) are being established worldwide in response to this.

Entomopathogenic fungi were the first microorganisms identified as insect pathogens, whereas entomopathogenic bacteria were the first to be commercialized (Lacey et al., 2001). Lecanicillium lecanii (=Verticillium lecanii (Zimmerman)Viegas) belongs to Deuteromycotina, Hyphomycetes, Moniliales, Moniliaceae, that is widely use entomopathogenic fungi in bio-control up to now. And the entomopathogenic fungal species described and commercialized, L. lecanii (Zare & Gams, 2001) deserves further consideration as a broad range commercial biopesticide, due to its wide range of hosts and wide geographical distribution (Xie et al., 2015). Indeed, this species can infect the diamondback moth Plutella xylostella (L.) (Lepidoptera: Plutellidae) (Ravindran et al., 2018), aphids (Hemiptera: Aphididae) (Askary, Benhamou & Brodeur, 1999), the citrus mealybug Planococcus citri Risso (Hemiptera: Pseudococcidae) (Ghaffari et al., 2017), and the soybean cyst nematode Heterodera glycines Ichinohe (Tylenchida: Heteroceridae) (Shinya et al., 2008), and has also been documented to infect Bemisia tabaci (Zhu & Kim, 2011). In insects, the spores of entomopathogenic fungi germinate, and the fungal hyphae penetrate the epidermis and invade the tissues and organs until reaching the hemocoel (Duan et al., 2017). When the hyphae come into contact with the hemolymph, the defense system of the insects, which includes detoxifying and protective enzymes, is induced (Liu et al., 2013).

Physiological and biochemical approaches have been used to describe the chronological events leading to fungal infestation success in an insect host. Reactive oxygen species (ROS) are forms of atmospheric oxygen (Tian et al., 2016b) produced in the mitochondria that are equilibrated by cellular antioxidative mechanisms (Esmail et al., 2018). In many instances, microbial pathogens are associated with an increase in ROS, which induces an oxidative stress response in the host (Foyer & Noctor, 2013). The antioxidative mechanism of the cells includes antioxidant enzymes, such as catalase (CAT), superoxide dismutase (SOD), and peroxidase (POD), which degrade H2O2 to reduce oxidative damage (Felton & Summers, 1995). In addition to this antioxidative mechanism, insects also harbor detoxifying enzymes, such as carboxylesterase (CarE), glutathione-S transferase (GST), and acetylcholinesterase (AchE), which are able to metabolize exogenous toxicants (Xu et al., 2006), and have been the target of insecticide synergist research (Wang et al., 2016). The effects of these insect detoxifying enzymes in response to the fungal entomopathogen L. lecanii in the spiraling whitefly Aleurodicus dispersus Russell (Hemiptera: Aleyrodidae) have recently been demonstrated (Liu et al., 2013). These changes in defensive enzymes are deserving further attention, due to its practical considerations.

Due to the lack of studies and the economic importance of Bemisia tabaci, the objective of this study was to determine the pathogenic effect of L. lecanii strain JMC-01 at the nymphal stages of Bemisia tabaci by evaluating the disruption of immune mechanisms.

Materials and Methods

Entomopathogen strain and insect collection

Lecanicillium lecanii strain: the L. lecanii strain JMC-01 was isolated from Bemisia tabaci infected nymphs from a greenhouse in Yinchuan, Ningxia (38°33′N, 106°08′E), China in May 2017. The JMC-01 strain was deposited at the China Center for Type Culture Collection with the accession number M2018303. The strain status was determined based on ITS sequence divergence to the reference strain (Jiang, 2018). The JMC-01 strain reference ITS nucleotide sequence was deposited in GenBank with the identification number MH312006.

Insect: the whitefly Bemisia tabaci Q-biotype was collected from a tomato greenhouse in Yinchuan, Ningxia (38°33′N, 106°08′E) in July 2018. Biotype assignment was performed as previously described (Gao, 2018). The tomato cultivar Bijiao was planted in a greenhouse in Yinchuan, Ningxia (38°33′N, 106°08′E) and cultivated using drip irrigation technology. Tomato was used as the host plant for two generations, following which the synchronized nymphs were collected for experimentation.

Preparation of the L. lecanii JMC-01 conidial suspension

The L. lecanii JMC-01 strain was inoculated on potato dextrose agar (PDA) plates, at 28 °C with a 12:12 (L:D, light:dark) photoperiod for 7 days (MJ-250 Mould Incubator; Jiangsu Zhengji Instruments Co. Ltd., Jiangsu, China). Spore suspensions were prepared by recovering the conidia from the PDA plates with a 0.05% Tween-80 solution. The solution was filtered with sterile cheesecloth to eliminate the hyphae, following which the concentration was adjusted to 1.0 × 108 conidia/mL with sterile water using a hemocytometer (Qiujing, Shanghai, China).

Bemisia tabaci nymph mortality induced by L. lecanii JMC-01

Tomato leaves with 1st-, 2nd-, 3rd- or 4th-instar nymphs (only one leaf was selected for each instar nymph) were immersed in L. lecanii JMC-01 solution at 1.0 × 108 conidia/mL for, 30 s or in a control solution of 0.05% Tween-80. After immersion, each leaf was sealed in a standard Petri dish, with its petiole wrapped in a moistened cotton ball. The plates were incubated in an artificial climate chamber (RQX-250; Shanghai Yuejin Medical Devices Co., Ltd., Shanghai, China) at 28 ± 2 °C, 70% ± 10% RH, and 12:12 (L:D) photoperiod. There were three replicates per treatment. Deaths were recorded daily, and the cumulative corrected mortality was calculated as follows: Accumulative corrected mortality (%)=Infection mortality−Control mortality1−Control mortality×100%

Susceptibility of 3rd-instar Bemisia tabaci nymphs to different JMC-01 concentrations

The L. lecanii JMC-01 suspensions were prepared as described above at different conidial concentrations: 1 × 108, 1 × 107, 1 × 106, 1 × 105, and 1 × 104 conidia/mL.

Three tomato leaves with 3rd-instar Bemisia tabaci nymphs were immersed for, 30 s at each JMC-01 test concentration, and the leaves were incubated as described above. Deaths were recorded on a daily basis, and were used to determine the cumulative corrected mortality for each conidial concentration.

Protective and detoxifying enzyme activity determination

Tomato leaves with 3rd-instar Bemisia tabaci nymphs were infected with L. lecanii JMC-01 at 1 × 108 conidia/mL, using the immersion procedure described above. Treated and control (0.05% Tween-80) leaves were immersed in L. lecanii JMC-01 solution at 1.0 × 108 conidia/mL for 30 s. After immersion, each leaf was sealed in a standard Petri dish, with its petiole wrapped in a moistened cotton ball. The plates were incubated in an artificial climate chamber (RQX-250; Shanghai Yuejin Medical Devices Co., Ltd., Shanghai, China) at 28 ± 2 °C, 70% ± 10% RH, and 12:12 (L:D) photoperiod.

Sample processing: the animal tissue, was weighted and nine-times the volume of normal saline by weight was added (weight (g):volume (mL) = 1:9), the samples were then ground with liquid nitrogen to make a 10% tissue homogenate, which was then centrifuged at 2,500 rpm for 10 min (Sigma D-37520; Sigma-Aldrich; Nanjing Beiden Medical Co., Ltd. Nanjing, China). The supernatant was then diluted to 1% tissue homogenate with normal saline for experimentation.

Protein content determination

The 563 μg/mL standard solution, working fluid, stop application solution and normal saline were purchased from the Jian Cheng Bioengineering Institute (Nanjing, China).

After combining the solutions, they were placed at room temperature for 5 min, and measured colorimetrically at 562 nm (L5S UV Spectrophotometer; Shanghai Yidian Analytical Instrument Co., Ltd., Shanghai, China) (Table 1). Double-distilled water served as the blank control.

Table 1 The steps of protein content determination.

	Blank tube	Standard tube	Measuring tube	
Double distilled water (μL)	20			
563 μg/mL standard solution (μL)		20		
Sample (μL)			20	
Working fluid (μL)	250	250	250	
Mix, set at 37 °C water bath for 30 min (digital thermostat water bath)	
Stop application solution (μL)	750	750	750	

The protein concentration was determined as follows: Protein(μgprot/mL)=Measure OD−Blank ODStandard OD−Blank OD×Standard solution (563μg/mL)                                                            ×Sample dilution before determination

SOD activity determination

Reagent one application solution, reagent two solution, reagent three solution, reagent four application solution, chromogen solution, and normal saline were purchased from the Jian Cheng Bioengineering Institute, Nanjing, China.

After combining the solutions, they were placed at room temperature for 10 min, and measured colorimetrically at 550 nm (Table 2). Double-distilled water served as the blank control.

Table 2 The steps of SOD activity determination.

Reagent	Measuring tube	Control tube	
Reagent one application solution (mL)	1.0	1.0	
Sample (mL)	0.1		
Double distilled water (mL)		0.1	
Reagent two solution (mL)	0.1	0.1	
Reagent three solution (mL)	0.1	0.1	
Reagent four application solution (mL)	0.1	0.1	
Mix, set at 37 °C water bath for 40 min (digital thermostat water bath)	
Chromogen solution (mL)	2	2	

Superoxide dismutase activity was determined as follows: SOD(U/mg prot)=Control OD−Measure ODControl OD÷50%                                                   ×Total volume of reaction solutionSample size (mL)                                                   ÷Protein concentration of the sample to be tested (mg prot/mL)

POD activity determination

Reagent one solution, reagent two application solution, reagent three application solution, reagent four solution, and normal saline were purchased from the Jian Cheng Bioengineering Institute, Nanjing, China.

The solutions were combined and centrifuged at 3,500 rpm for 10 min (Sigma D-37520; Sigma-Aldrich, Germany), following which the supernatant was measured colorimetrically at 420 nm (Table 3). Double-distilled water served as the blank control.

Table 3 The steps of POD activity determination.

	Blank tube	Measuring tube	
Reagent one solution (mL)	2.4	2.4	
Reagent two application solution (mL)	0.3	0.3	
Reagent three application solution (mL)	0.2	0.2	
Double distilled water (mL)	0.1		
Sample (mL)		0.1	
Set at 37 °C water bath for 30 min (digital thermostat water bath)	
Reagent four (mL)	1.0	1.0	

Peroxidase activity was determined as follows: POD(U/mg prot)=Measure OD−Blank OD12×1×Total volume of reaction solutionSample size (mL)                                                   ÷Reaction time (30 min)                                                   ÷Protein concentration of the sample to be tested (mg prot/mL)                                                   ×1,000

CAT activity determination

Reagent one solution, reagent two solution, reagent three solution, reagent four solution, and normal saline were purchased from the Jian Cheng Bioengineering Institute, Nanjing, China.

After combining the solutions, they were measured colorimetrically at 405 nm (Table 4). Double-distilled water served as the blank control.

Table 4 The steps of CAT activity determination.

	Control tube	Measuring tube	
Sample (mL)		0.05	
Reagent one solution (37 °C preheat) (mL)	1.0	1.0	
Reagent two solution (37 °C preheat) (mL)	0.1	0.1	
Mix, set at 37 °C water bath for 1 min (digital thermostat water bath)	
Reagent three solution (mL)	1.0	1.0	
Reagent four solution (mL)	0.1	0.1	
Sample (mL)	0.05		

Catalase activity was determined as follows: CAT(U/mg prot)=(Control OD−Measure OD)×271×160×0.05                                                   ÷Protein concentration of the sample to be tested (mg prot/mL)

CarE activity determination

The working fluid and normal saline were purchased from the Jian Cheng Bioengineering Institute, Nanjing, China.

Sample processing: the sample processing was as described in the protein content determination step above, except that the tissue homogenate was centrifuged at 12,000 rpm for 4 min.

The steps were as follows: The spectrophotometer was preheated for at least 30 min and the wavelength was adjusted to 450 nm. The machine was blanked with double-distilled water.

The working fluid was preheated at 37 °C for at least 30 min.

Blank tube: Five μL of distilled water was added to a blank glass cuvette, to which 1,000 μL of preheated working solution was sequentially added to a one mL glass cuvette. The solution was rapidly mixed, and light absorption A1 and A2 was measured at 450 nm 10 and 190 s, ΔABlank tube = A2 − A1.

Measuring tube: Five μL of supernatant was sequentially added to a one ml glass cuvette, 1,000 μL of preheated working solution, and rapidly mixed, and light absorption of A3 and A4 were measured at 450 nm, ΔAMeasuring tube = A4 − A3. CarE(U/mg prot)=(ΔAMeasuring tube−ΔABlank tube)×V÷(Cpr×VSample)÷T

V: total volume of the reaction solution, 1.005 mL;

Cpr: protein concentration of the sample to be tested (mg prot/mL);

VSample: adding of supernatant volume to the reaction system (mL), 0.005 mL;

T: catalytic reaction time (min), 3 min.

AchE activity determination

One μmol/mL standard application solution, substrate buffer, chromogen application solution, inhibitor solution, transparent solution, and normal saline were purchased from the Jian Cheng Bioengineering Institute, Nanjing, China.

After combining the solutions, they were placed at room temperature for 15 min and measured colorimetrically at 412 nm (Table 5). Double-distilled water served as the blank control.

Table 5 The steps of AchE activity determination.

	Measuring tube	Control tube	Standard tube	Blank tube	
Sample (mL)	0.1				
One μmol/mL standard application solution (mL)			0.1		
Double distilled water (mL)				0.1	
Substrate buffer (mL)	0.5	0.5	0.5	0.5	
Chromogen application solution (mL)	0.5	0.5	0.5	0.5	
Mix, set at 37 °C water bath for 6 min (digital thermostat water bath)	
Inhibitor solution (mL)	0.03	0.03	0.03	0.03	
Transparent solution (mL)	0.1	0.1	0.1	0.1	
Sample (mL)		0.1			
Note:

The same sample was added to the control tube and the measuring tube, but the order was different. The blank tube was not sampled and distilled water was used instead of the sample.

AchE activity was determined as follows: AchE(U/mg prot)=Measure OD−Control ODStandard OD−Blank OD×Standard concentration (1 μmol/mL)                                                    ÷Protein concentration of the sample to be tested (mg prot/mL)

GST activity determination

Matrix fluid, reagent two application solution, anhydrous alcohol, GSH standard application solution, 20 μmol/mL GSH standard solution, reagent three application solution, reagent four application solution, and normal saline were purchased from the Jian Cheng Bioengineering Institute, Nanjing, China.

Enzymatic reaction:

The solutions were combined and centrifuged at 3,500 rpm for 10 min (Sigma D-37520; Sigma-Aldrich, Germany) (Table 6). The supernatant was then used in the chromogen reaction.

Table 6 The steps of enzymatic reaction.

	Measuring tube	Control tube	
Matrix fluid (mL)	0.3	0.3	
Sample (mL)	0.1		
Mix, set at 37 °C water bath for 10 min (digital thermostat water bath)	
Reagent two application solution (mL)	1	1	
Anhydrous alcohol (mL)	1	1	
Sample (mL)		0.1	

Chromogen reaction:

The solutions were combined and placed at room temperature for 15 min, following which they were measured colorimetrically at 412 nm (Table 7). Double-distilled water served as the blank control.

Table 7 The steps of chromogen reaction.

	Blank tube	Standard tube	Measuring tube	Control tube	
GSH standard application solution (mL)	2				
20 μmol/mL GSH standard solution (mL)		2			
Supernatant (mL)			2	2	
Reagent three application solution (mL)	2	2	2	2	
Reagent four application solution (mL)	0.5	0.5	0.5	0.5	
Note:

The same sample was added to the control tube and the measuring tube, but the order was different. The blank tube was not sampled and distilled water was used instead of the sample.

Glutathione-S transferase activity was determined as follows: GST(U/mg prot)=Control OD−Messure ODStrandard OD−Blank OD×Standard concentration 20 μmol/mL                                                   ×Reaction system dilution factor (6 times)÷Reaction time (10 min)                                                   ÷[Sample volume (0.1 mL)                                                   ×Protein concentration of the sample to be tested (mg prot/mL)]

Determination of weight, and water and fat content of the Bemisia tabaci nymphs after infestation with L. lecanii JMC-01

Tomato leaves with 3rd-instar Bemisia tabaci nymphs exposed to 1.0 × 108 conidia/mL or the control treatment (0.05% Tween-80). The treated and control leaves were placed in similar Petri dishes. The same Petri dish method as above was then used.

The treatment and control group were selected one hundred 3rd-instar nymphs for experimentation after 0, 12, 24, 36, 48, 60, and 72 h, respectively. First determining the total fresh weight of 100 nymphs prior to infection (Mettler Toledo LE204E/02 electronic balance), the nymphs were dried by placing each batch at 60 °C for 48 h in an electrothermal blowing dry box (Shanghai Yiheng Technology Co., Ltd., Shanghai, China), and weighed in a similar method as for the determination of dry weight (dry mass, DM).

Water content (WC) was determined using the formula WC = (FW − DM)/FW × 100%, where DW is the dry mass determined as explained above, and FW is the fresh weight determined as above.

Lipid extraction was performed with the dried nymphs. The dried nymphs were grinded under liquid nitrogen in a centrifuge tube. One mL of chloroform isoamyl alcohol (24:1) and 0.5 mL of methanol (99.99%) was added to each tube, mixed, and then centrifuged at 4,500 rpm for 10 min. The supernatant was discarded. The precipitate was extracted again with one mL of chloroform isoamyl alcohol (24:1) and 0.5 mL of methanol (99.99%) by centrifugation at 4,500 rpm for 10 min. The final remaining precipitate was dried in an oven at 60 °C for 48 h to determine the constant dry mass (LDM).

Fat content (FC) was determined using the formula FC = (DM − LDM)/DM × 100%, where DM is the dry mass determined as explained above, and LDM is the constant dry mass determined after lipid extraction.

There were three replicates per treatment and time point, and 100 nymphs per replicate.

Data analysis

Excel 2010 (Microsoft Corporation, Albuquerque, NM, USA) was used to process all the data. All results are expressed as the mean ± standard deviation. Statistical analysis of the data was performed using one-way analysis of variance with SPSS version 21.0 (SPSS; IBM Corp., Armonk, NY, USA). Multiple comparisons of the means were performed using Duncan’s (D) tests at a significance level of P = 0.05. All figures were produced using Origin 8.0.

Results

Morphological characteristics of the Bemisia tabaci nymphs

Figure 1 shows the morphological characteristics of Bemisia tabaci under L. lecanii JMC-01 infection as observed under a microscope (Leica Microsystems Wetzlar GmbH, Wetzlar, Germany). The surface is covered with hyphae.

Figure 1 Morphological characteristics of the B. tabaci nymph induced by L. lecanii JMC-01.

Mortality of the Bemisia tabaci nymphs

Figure 2 indicates the cumulative mortality induced by L. lecanii JMC-01 to each Bemisia tabaci immature stage. The cumulative corrected mortality of the nymph instars was as follows (from high to low): 2nd instar > 3rd instar > 1st instar > 4th instar > egg. The 2nd- and 3rd-instar nymphs were most affected, with corrected cumulative mortality percentages of 82.22% and 75.55%, respectively.

Figure 2 Cumulative corrected mortality of L. lecanii JMC-01 infestation on B. tabaci nymphs.

Times marked with different uppercase letters on the same line are significantly different (P < 0.05). The different lowercase letters indicate significant differences between the treatment and control groups (P < 0.05) at the same time point.

The initial dose of L. lecanii JMC-01 affects the 3rd-instar Bemisia tabaci nymphs

As indicated in Fig. 3, increasing doses of L. lecanii JMC-01 (from 1 × 104 to 1 × 108 conidia/mL) also increased the corrected cumulative mortality of the 3rd-instar nymphs, reaching a maximum of 75.55% at 1 × 108 conidia/mL after 6 days.

Figure 3 Cumulative corrected mortality of the 3rd-instar B. tabaci nymphs following exposure to different concentrations of L. lecanii JMC-01.

Each data point indicates the corrected cumulative mortality for each time period.

Protective and detoxifying enzyme activity determination

The highest activity of SOD (43 U/mg prot) was detected on the 2nd day, reaching 1.1-fold that of the control (Fig. 4). The highest activities of POD and CAT were 20 and 6.3 U/mg prot on the 3rd day, respectively, and reached 2.4- and 1.4-fold that of the control level (Figs. 5 and 6). Following this, the activities of protective enzymes decreased. The lowest activities of SOD, POD, and CAT were 30, 8.5, and 1.3 U/mg prot on the 5th day, respectively (Figs. 4–6).

Figure 4 Effects of SOD activities of the 3rd-instar B. tabaci nymphs infested with L. lecanii JMC-01.

Times marked with different uppercase letters on the same line are significantly different (P < 0.05). The different lowercase letters indicate significant differences between the treatment and control groups (P < 0.05) at the same time point.

Figure 5 Effects of POD activities of the 3rd-instar B. tabaci nymphs infested with L. lecanii JMC-01.

Times marked with different uppercase letters on the same line are significantly different (P < 0.05). The different lowercase letters indicate significant differences between the treatment and control groups (P < 0.05) at the same time point.

Figure 6 Effects of CAT activities of the 3rd-instar B. tabaci nymphs infested with L. lecanii JMC-01.

Times marked with different uppercase letters on the same line are significantly different (P < 0.05). The different lowercase letters indicate significant differences between the treatment and control groups (P < 0.05) at the same time point.

The highest activities of CarE and AchE were 10.5 and 0.32 U/mg prot. These levels were observed on the 3rd day and were 2.2- and 4.3-fold that of the control level, respectively (Figs. 7 and 8). The highest GST activity was 64 U/mg prot on the 2nd day and was 2.7-fold that of the control level (Fig. 9). After the 3rd day, the activities of detoxifying enzymes decreased, and the lowest activities of CarE, AchE, and GST, respectively, reached 3.5, 15, and 0.05 U/mg prot on the 5th day (Figs. 7–9).

Figure 7 Effects of CarE activities of the 3rd instar nymph of B. tabaci infested with L. lecanii JMC-01.

Times marked with different uppercase letters on the same line are significantly different (P < 0.05). The different lowercase letters indicate significant differences between the treatment and control groups (P < 0.05) at the same time point.

Figure 8 Effects of AchE activities of the 3rd instar nymph of B. tabaci infested with L. lecanii JMC-01.

Times marked with different uppercase letters on the same line are significantly different (P < 0.05). The different lowercase letters indicate significant differences between the treatment and control groups (P < 0.05) at the same time point.

Figure 9 Effects of GST activities of the 3rd instar nymph of B. tabaci infested with L. lecanii JMC-01.

Times marked with different uppercase letters on the same line are significantly different (P < 0.05). The different lowercase letters indicate significant differences between the treatment and control groups (P < 0.05) at the same time point.

Determination of the weight and water and fat contents of the of Bemisia tabaci nymphs

The lowest changes in weight were observed at 24–36 h. At 72 h, the weight of the infected group was 0.78-fold that of the control (Fig. 10).

Figure 10 Changes in weight of the 3rd instar B. tabaci nymphs infected with L. lecanii JMC-01.

Times marked with different uppercase letters on the same line are significantly different (P < 0.05). The different lowercase letters indicate significant differences between the treatment and control groups (P < 0.05) at the same time point.

The WC of Bemisia tabaci continuously decreased after infection with L. lecanii. At 72 h, the WCs of the infected and control groups were lowest reaching 56% and 66%, respectively (Fig. 11).

Figure 11 Changes in water content of the 3rd instar B. tabaci nymphs infected with L. lecanii JMC-01.

Times marked with different uppercase letters on the same line are significantly different (P < 0.05). The different lowercase letters indicate significant differences between the treatment and control groups (P < 0.05) at the same time point.

Until 36 h after infection, the changes in FC were not significantly different from the control level. At 72 h, the FC of the infected and control groups was the lowest, reaching 13% and 20.5%, respectively (Fig. 12).

Figure 12 Changes in and fat content of the 3rd instar B. tabaci nymphs infected with L. lecanii JMC-01.

Times marked with different uppercase letters on the same line are significantly different (P < 0.05). The different lowercase letters indicate significant differences between the treatment and control groups (P < 0.05) at the same time point.

Discussion

The fungus penetrated the insect epidermis via the germ tubes and appressoria, following which the conidia invaded the nymphs and began to enter the hemocoel. Ultimately, the hyphae covered the host surface and had colonized the body cavity (Zhou et al., 2017). Previously, L. lecanii caused over 90% mortality of vegetable pest, such as aphids, Plutella xylostella (Ravindran et al., 2018; Saruhan, 2018; Sugimoto et al., 2003). In this study, mortality increased greatly during the first 5 days of infection, with the maximum mortality is 82.22% being reached on the 6th day. Accordingly, the activities of detoxification and protective enzymes were lowest on the 5th day, indicating that as the infected nymphs of Bemisia tabaci neared death on the 5th day, their enzyme activity was reduced.

Insects are protected from the stresses of adverse conditions by various physical barriers, including a cuticular exoskeleton, peritrophic membrane, and an immune system that reduces pathogen infection (Chen & Lu, 2017). These fungi stimulate the stress responses of the insect detoxification system and the protective enzyme system under adverse conditions by changing the function of ion channels (Zhang et al., 2017). The major components of the antioxidant defense system of insects include the antioxidant enzymes SOD, CAT, and POD (Li et al., 2016b). When insects are stimulated by exogenous compounds, SOD converts the superoxide radical O2 into H2O2. Then, POD and CAT convert the H2O2 into H2O. The imbalance between oxidative stress and antioxidant responses contributes to disease and the death of insect hosts (Felton & Summers, 1995).

Our study showed that, after infection of Bemisia tabaci by L. lecanii, the activities of SOD, CAT, and POD initially increased but then decreased thereafter, and the maximum activities protective enzymes were observed on the 2nd day or 3rd day. Previous studies (Yang et al., 2015; Ye et al., 2018; Zhou et al., 2017) indicated agricultural insects infested by entomogenous fungus, the activities of SOD, CAT, and POD initially increased but then decreased. The increased activity of SOD, CAT, and POD effectively preventing the formation of more toxic substances such as hydroxyl radicals and helped increase the resistance of Bemisia tabaci (Ding, Zhang & Chi, 2015). Under L. lecanii infection, ROS and other toxic substances stimulated an immune system response in Bemisia tabaci (Li et al., 2016a). To resist the adverse environmental influence and maintain normal physiological activities, the enzyme activities sharply increased. However, the internal spread of the pathogen led to the destruction of the internal tissue structure of the insect and subsequent collapse of the immune system. In addition, the ROS scavenging system might not have been able to remove the excessive quantity of free radicals, leading to reduced enzyme activity and the death of the insect (Li et al., 2016a). So, the activities of SOD, CAT, and POD were decreased on the 5th day. GSTs participate in detoxification metabolism and and catalyze a combination of toxic substances with glutathione and also promote the excretion of toxic chemicals and pathogenic substances (Mathews et al., 2002; Schama et al., 2016). CarEs can catalyze the hydrolysis of ester bonds, and their major physiological functions include lipid metabolism, detoxification metabolism of exogenous compounds and biochemical regulatory functions (Guo et al., 2015). AchE is a target for organophosphorus and carbamate insecticides (Ding et al., 2001). Some exogenous compounds, such as pesticides and pathogenic fungi, can be altered by insect detoxification enzymes. This suggests that L. lecanii can promote the detoxification metabolism of Bemisia tabaci, which is beneficial for the discharge of exogenous toxicants. With the increase in the level of Bemisia tabaci infection with L. lecanii exposure time, the exogenous toxicants overpowered the detoxification metabolism, resulting in the eventual reduction in enzyme activities, and ultimately, insect death. We found that the activities of CarEs, AchE, and GST initially increased but then decreased, and the maximum activities of CarEs, AchE, and GST were observed on the 2nd day or 3rd day after infection. Effects of Isaria fumosorosea infection on different enzyme activities in the adult in vivo of Bemisia tabaci indicated that the maximum activities of GSTs and CarE were observed on the 48–60 h (Tian, Diao & Ma, 2016a). Besides, these findings are similar to previous study (Liu et al., 2013; Zhang et al., 2015). Insects infested with entomopathogenic fungi initially exhibit elevated enzyme activities that decline as the fungal infection continues (Tian, Diao & Ma, 2016a). The entomopathogenic fungus L. lecanii can be used to control Bemisia tabaci nymphs, but the prevention and control effect is slower than with chemical insecticides. However, the use of entomopathogenic fungi in combination with insecticides to control pests could increase their speed and efficacy (Purwar & Sachan, 2006).

A study of the pathogenicity and control potential of Beauveria bassiana on the onion fly showed that the weight increment was smallest after 48 h. The WC and FC continued to decrease, and the WC and FC of the infection level were 0.81- and 0.69-fold that of the control level, respectively, at 72 h (Zhang, 2017). Screening of the strains of the highly pathogenic Beauveria bassiana on soybean pod borers and the assessment biophysical and biochemical effects on their hosts indicated reductions in weight, WC, and FC (Tian, 2014). These studies corroborate our findings. In the present study, Bemisia tabaci nymphs infested with L. lecanii JMC-01 gradually lost vitality until death. This process causes many physiological changes in the insects. Thus, L. lecanii could constitute a useful alternative biopesticide for Bemisia tabaci population management. Biocontrol can reduce insecticide resistance and increase environmental and non-target organism safety.

Conclusions

We observed that L. lecanii JMC-01 affected the viability of the B. tabaci Q-biotype, by inducing mortality, affecting the activities of protective and detoxifying enzymes, and by significantly reducing the weight, and water and FC. Thus, L. lecanii impacted the physiological functioning of B. tabaci by directly acting on molecular targets and by indirectly acting on detoxification and protective enzymes (Bantz et al., 2018). These results indicate that this fungal strain could constitute an effective biological control for B. tabaci in agriculture.

Supplemental Information

Supplemental Information 1 Corrected cumulative corrected mortality of L. lecanii JMC-01 infestation on B. tabaci nymphs.

Raw data for Figs. 2 and 3 for the time period of 1–6 days.

Click here for additional data file.

Supplemental Information 2 Effects of SOD, POD, and CAT activities of the 3rd-instar B. tabaci nymphs infested with L. lecanii JMC-01.

Raw data for Figs. 4–6 for the time period of 1–5 days.

Click here for additional data file.

Supplemental Information 3 Effects of CarE, AchE, and GST activities of the 3rd instar nymph of B. tabaci infested with L. lecanii JMC-01.

Raw data for Figs. 7–9 for the time period of 1–5 days.

Click here for additional data file.

Supplemental Information 4 Changes in weight, water, and fat content of the 3rd instar B. tabaci nymphs infected with L. lecanii JMC-01.

Raw data for Figs. 10–12 for the time period of 0–72 h.

Click here for additional data file.

We are grateful to Master Kai Gao for providing the biotype of B. tabaci and we to Master Hui Wang for helping with the formatting of this paper. We are grateful to two anonymous reviewers and Jasmine Janes for their comments on an early version of the manuscript. We thank LetPub for its linguistic assistance during the preparation of this manuscript.

Additional Information and Declarations

Competing Interests

Author Contributions

Field Study Permissions

Patent Disclosures

DNA Deposition

Data Availability

The authors declare that they have no competing interests.

The strain Lecanicillium lecanii JMC-01 is in the process of patenting (Patent Number: 201811310451.X), but the patent is currently in the trial stage.

Ting Xie conceived and designed the experiments, performed the experiments, analyzed the data, prepared figures and/or tables, authored or reviewed drafts of the paper, approved the final draft.

Ling Jiang performed the experiments, analyzed the data, prepared figures and/or tables.

Jianshe Li performed the experiments, authored or reviewed drafts of the paper, approved the final draft.

Bo Hong performed the experiments, authored or reviewed drafts of the paper, approved the final draft.

Xinpu Wang performed the experiments, contributed reagents/materials/analysis tools, authored or reviewed drafts of the paper, approved the final draft.

Yanxia Jia conceived and designed the experiments, performed the experiments, analyzed the data, contributed reagents/materials/analysis tools, authored or reviewed drafts of the paper, approved the final draft.

The following information was supplied relating to field study approvals (i.e., approving body and any reference numbers):

Shuangyu Zhang granted access to his private greenhouse for the duration of the experiments.

The following patent dependencies were disclosed by the authors:

The strain Lecanicillium lecanii JMC-01 has been deposited in China Center for Type Culture Collection on 2018.06.05 and is being patented (Patent Number: 201811310451.X), but the patent is in the trial stage.

The following information was supplied regarding the deposition of DNA sequences:

The JMC-01 strain reference ITS nucleotide sequence is available at GenBank: MH312006.

The following information was supplied regarding data availability:

The raw measurements for Figs. 2–12 are available as a Supplemental File.

The strain Lecanicillium lecanii JMC-01 is available in the China Center for Type Culture Collection (No: M2018303).

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
