# Peer review of "Effects of Lecanicillium lecanii strain JMC-01 on the physiology, biochemistry, and mortality of Bemisia tabaci Q-biotype nymphs"

_PeerJ, doi:10.7717/peerj.7690_

## Round 0.1 · original submission · Major Revisions

The manuscript presents data that could be useful in biocontrol studies. However, there are several areas that require further attention. The methods are not fully detailed thus, few people could attempt to replicate this work or use this manuscript for methodological inspiration. The introduction and background in the abstract are lacking sufficient information about the study species and relationships. At present the discussion is poorly constructed. It provides several results (which should be moved to the results) and doesn't really compare or contrast effectively with the existing literature. Also, there is a very lengthy section pertaining to the enzyme activity but very little discussion of any other results. I suggest re-working these sections to improve detail, clarity and flow.

Lines 12-15 – the background provided is exceptionally brief for readers unfamiliar with this system. What does the fungus (which has undergone a genus-level name change) have to do with the insect?
Line 52 – Use the full genus name when starting a sentence
Line 52-53 – citation?
Line 67-69 – citation?
Line 74 – what is this species? How is it relevant? Family and authority names for all species should be provided at their first use.
Line 91 – cite the previous study
Line 114 - Use the full genus name when starting a sentence
Line 123-126 – what about all these enzymes? How was the enzyme activity measured?
Line 131 – how was the fresh weight determined? Did you weigh all nymphs prior to infection?
Line 150 – remove this first sentence
Line 151 – period after reference to figure
Line 172 – merge with previous paragraph/sentence
Line 182 – merge with previous paragraph/sentence
Line 187-189 – these seem like results
Line 189-195 – these statements need more context. What is the point of telling the reader about all these other species and isolates? Is it to compare and contrast, or is it simply listing facts?
Line 211 – new paragraph
Line 213 – preventing rather than prevented
Line 214-216 – how is it consistent with the other studies? What did they show exactly?

Figures – I find the labelling of points here to be confusing. For example, does the plot indicate that the bottom ‘egg’ line was ‘non-significant’ between time point 4, 5 and 6? Does that also mean that these points were ‘non-significant’ from the control because they have the same lowercase letter?

[]

Reviewer 1 ·

Basic reporting

Even if the authors state that they have used an English grammar and style proof-reading service, the actual manuscript is hard to understand. Specially at the Material and Methods section, which in some parts lack of complete sense.
A part from this, I thought that the main objective was achieved, as they want to indicate the performance of a new fungus strain which could be used in biological control of B. tabaci. But the authors lack clarity, and they have promising results, that would benefit from a profesional writer to increase its impact.
Even if they have an appropriate number of references, they are missing one important point, that is the taxonomical position of the fungus species they were using. Some other references are missing at the introduction part, that could be also useful at the discussion.
The figure listing should be checked as in the pdf file the figures are listed from 1 to 11, whereas in the main text, only reach to fig. 6.

Experimental design

The authors have stated a research gap, but they have failed in describing the most important section at the Material & Methods.
I tried to re-write this section, but the exact protocols used with the corresponding references which are missing, some of these are indicated in the attached file.

My main concern is that the special focus that the authors state that have this manuscript is missing, ie. The authors focus that is of special importance the study of protective and detoxifying enzyme activities, but they don´t describe how this has been carried out. The authors state in lines 510-516 :
“Tomato leaves with 3rd instar B. tabaci nymphs were immersed in a conidia suspension of V. lecanii JMC-01(1×108conidia/mL). After this method was used, the same petri dish method as above was then used. The temperature was 20°C in the laboratory.
The protective enzyme activities of superoxide dismutase (SOD), catalase (CAT) and peroxidase (POD), and the detoxifying enzyme activities of carboxylesterase (CarE), glutathione-S transferase (GSTs) and acetylcholinesterase (AchE) from Jian Cheng Bioengineering Institute, Nanjing.”

To me, this last sentence is understandable. Where are the methods? how the activities performed by each of these enzymes were measured?
This is an example, the following sub-section on weight, water and fat content, also have many missing points.

Validity of the findings

The presented data seems robust and statistically sound, but the authors have not squeezed them in full.
I thought that the authors could find more on their results, as finding a relationship between conidia concentration and mortality index, not sure if they could find a linear or if they should use a more complicated method (like the GLM). The results they provide with this part, deserves this further step, I thought that the manuscript will be really improved, adding a point not presented by any other one.
The conclusions are clearly linked to the supporting results, I have just edited them to give a more appropriate style, but it's up to the authors if they would like to return to the original version.

Additional comments

The manuscript describes the biological control power of a new fungus strain against one of the major worldwide agricultural pests, the whitefly Bemisia tabaci.
It is a great job, but deserves more work on it, specially done by an English native speaker, to polish the written style. Apart from that, I found it really interesting, including the description of a new fungus strain that the authors or their institution could patent.

Annotated reviews are not available for download in order to protect the identity of reviewers who chose to remain anonymous.

Reviewer 2 ·

Basic reporting

Recently, entomopathogenic fungi as myco-biocontrol resource offer an attractive alternative to the use of integrated pest management or chemical pesticides. In line with this, Ting Xie et al. attempted to investigate the effects of Verticillium lecanii strain JMC-01 on the physiology, biochemistry, and mortality of Bemisia tabaci Q biotype nymphs. Overall, they provided some evidence showing host physiological reactions to infection with V. lecanii JMC-01. However, the manuscript should not only be thoroughly improved in writing with a revision by a native speaker, but the structure should also be modified, and additional data should be provided before considering for publication.

In the introduction, the classification of Verticillium lecanii (Lecanicillium lecanii) should be clearly described. Also, it would be necessary to provide a more detailed description regarding a potential impact on biological control by its validated role of toxicity against insect host, for example, the results from a study of Ravindran et al. (2018) mentioned in this manuscript.

In the Materials and methods section, how did the authors select B. tabaci nymphs infected by V. lecanii JMC-01 for further analyses at 0 h, 12 h, 24 h, 36 h, 48 h, 60 h, and 72 h? Which method was used to determine water content in this study?

Because V. lecanii JMC-01 was isolated from the nymphs of B. tabaci and used as a novel isolate in this study, the results section should be included characteristics of the B. tabaci following infection with V. lecanii MC-01. MH312006 can be trackable, although the author and information were not exactly as shown in this manuscript. A more detailed and informative description representing obtained data should be added, for example, the statement in line 149-156 was largely uncomplemented. In addition, all figure legends should be revised with a brief description of methods. Figures should be precisely represented in the text, therefore figure 3-10 should be reconstructed. When comparison, the result for each time point should be used. For example, the statement in line 170-171 was incorrect (~2.7 fold for day 2 instead of 4.5 fold as mentioned in the manuscript).

The references contain many typos and are still in an inconsistent format. Moreover, all of the journals were lacking. Reference of Ravindran et al. (2018) was represented two times.

Experimental design

no comment

Validity of the findings

no comment

---

## Round 0.2 · Minor Revisions

The authors have made significant improvements based on the previous review comments. At this point, I advise the authors to seek the assistance of an editing service (please note that PeerJ does offer such services but you may choose another). There are numerous grammatical issues, particularly within the introduction and discussion. Resolving these issues would greatly improve the manuscript and ensure good readership. My biggest concern at the moment is that a reader may not understand all of the content and impact based on these grammatical issues, which would be a disservice to your hard work and communication.

---

## Round 0.3 · Minor Revisions

The manuscript is making progress. However, there are several areas that require more clarification before the manuscript can be published.

Line 18 – change to insects suffering from disease
Line 93 – delete accounting

Line 107-108 – can you explain why it says third instar larvae were collected for future experimentation but later on line 119 it says leaves with first, second or third instar larvae were used?

Line 145 – what is the specific method? Is it the protein and SOD determination? Using a colon suggests that something is following, perhaps a list.
Line 149-153 – these sentences would be better under the protective and detoxifying section so that they apply to all of the following sections. This would allow you to remove all of the sentences about sample processing (lines 167-168, 180-181, 193-194, 229-230, 243-244).

Line 154 – remove this sentence and place a reference to the table after the first sentence on line 155. Please do this for all the subsequent references to the tables.

Line 290 – remove the sentence about endnote

Results section – please use the full word ‘figure’ at the beginning of sentences

Line 358 – It’s not clear what ‘indicated agricultural insects by entomogenous fungus’ means. Please clarify or reword this sentence.

Line 359-362 – are all of these sentences referring to results from the studies mentioned on line 358? How do you know these things occurred?
Line 382 – why are some of these words presented as nouns (i.e., capital letters)?

Tables – please provide more descriptive titles for the tables. For example, what is the difference between a control and blank tube? Is the control a replicate of one of the samples? It doesn’t appear so.
Tables – is there any way to condense or reduce the number of tables?

---

## Round 0.4 · Minor Revisions

The manuscript is being returned with minor revisions again because, in some instances it is unclear where or how the changes were made, and in others, the changes require further detail.

I strongly suggest that the authors stop using highlighting to indicate where changes have been made in a revised manuscript. Highlighting does not do a good job of indicating exactly how or what changed. Word has an official track-changes function that should be used. This function provides a clear road map of what was changed, how it was changed (e.g., a deletion vs a word change) and generally makes for a much smoother review process.

I also suggest that when uploading a rebuttal, only upload the comments relevant to this revision. It was quite confusing to open the rebuttal document and see numerous comments relating to previous versions of the manuscript.

From going back and forth between versions of the manuscript, it does seem that the majority of changes were made. However, further clarification is needed for the change to lines 146-150. The requested change has been made but it is not clear that this sample processing applies to X number of proceeding sections. Please make sure that it is easily understood that these methods apply to a certain number of subsections.

---

## Round 0.5 · accepted · Accept

Thank you for your careful corrections.

#